# Research on the Impacts of Heterogeneous Environmental Regulations on Green Productivity in China: The Moderating Roles of Technical Change and Efficiency Change

**DOI:** 10.3390/ijerph182111449

**Published:** 2021-10-30

**Authors:** Yanchao Feng, Yong Geng, Zhou Liang, Qiong Shen, Xiqiang Xia

**Affiliations:** 1Business School, Zhengzhou University, Zhengzhou 450001, China; m15002182995@163.com (Y.F.); shenqiong@zzu.edu.cn (Q.S.); xqxia@zzu.edu.cn (X.X.); 2School of Environmental Science and Engineering, Shanghai Jiao Tong University, Shanghai 200240, China; ygeng@sjtu.edu.cn; 3School of Economics and Management, Harbin Institute of Technology, Harbin 150001, China

**Keywords:** heterogeneous environmental regulations, green productivity, technical change, efficiency change

## Abstract

Due to the publicly owned resource attributes of the ecological environment, the treatment and governance of the environment should be guided by governments, which are mainly represented as environmental regulations. However, whether environmental regulations affect green productivity and what effects heterogeneous environmental regulations have on green productivity are still implicit. In addition, the moderating roles of technical change and efficiency change are ignored. To examine these issues, this study investigated the impacts of heterogeneous environmental regulations on green productivity and the moderating roles of technical change and efficiency change using the dynamic spatial Durbin model based on the panel data of 30 Chinese provinces from 2000 to 2018. The results show the following: compared with efficiency change, technical change has a stronger promotion effect on green productivity in China; considering the spatial spillover effects and the temporal lag effects of green productivity simultaneously, the negative path-dependent feature is not supported any longer, while the spatial spillover effect is still the power source for promoting green productivity in China; the moderating roles of technical change and efficiency change for the nexus between heterogeneous environmental regulations and green productivity in China are partly and conditionally supported at national and regional levels; the direct and indirect effects of heterogeneous environmental regulations on green productivity at the regional level have a feature of spatial heterogeneity. This study provides both theoretical and practical implications, in particular for China, to promote green productivity from the dual perspectives of space and time.

## 1. Introduction

In the process of industrialization and urbanization, the problems of environmental pollution, climate change, and resource exhaustion have emerged and become threats to human beings in China [1]. Against this background, to achieve a win–win in environmental protection and economic development, Chinese governments have issued diverse kinds of environmental regulations that aim to induce firms’ production decisions and behaviors including resource saving and waste and pollution reduction [2]. However, all these production decisions and behaviors inevitably need a huge number of investments, which could add to the costs of production and the operation of firms and thus slow down their productivity growth in the short run [3]. Therefore, the final effect of environmental regulations at the micro level depends on the intensity of the compliance cost and innovation compensation.

Traditionally, per capita gross domestic product (GDP) has been widely utilized to act as the proxy indicator of economic performance. However, the overemphasis on economic growth has led to the unsatisfied situation of “trade environmental pollution for economic development” [4]. With increasing attention paid to high-quality development in the new era, economic performance is not the only goal of local governments, and environment quality should be incorporated into the evaluation of development quality, which elicits the concept of green productivity [5]. Green productivity growth has been utilized as the proxy index to measure the achievements of green growth incorporating not only economic outcomes but also energy consumption and pollution emissions [6].

In addition, it should be noted that green productivity can be decomposed into two indicators, which are technical change and efficiency change, representing the shift in the production frontier between two periods and the change in technical efficiency between two periods, respectively [7]. However, there is little literature focusing on the moderating effects of technical change and efficiency change when analyzing green productivity. Thus, to dig out the influencing mechanism of environmental regulations on green productivity, considering the moderating effects of technical change and efficiency change is critical for understanding the macro impacts of environmental regulations and optimizing the design of environmental regulation tools for governments.

As more and more increasingly serious problems of environmental pollution emerge, plenty of studies have explored the effect of environmental regulations on green productivity. However, no unified conclusions have been achieved until now, which has not only resulted in a fragmented literature in this field but also delivered the inspiration for this study. For example, Li and Wu (2017) tested the impacts of environmental regulation policy on green productivity growth by using the data of OECD countries’ industrial sectors and found a positive relationship [8]. However, Peng (2020) revealed that green productivity could be improved by environmental regulation in the local region but could be inhibited by the weighted environmental regulation in adjacent regions [9]. Furthermore, it should be pointed out that environmental regulations could be largely divided into three kinds (i.e., command–control, market incentive and public participation), while no study has focused on the heterogeneous effects of them on green productivity in China, which leaves another opportunity and motivation of this study [10].

Hence, this study aims to explore the effects of heterogeneous environmental regulations on green productivity, the moderating roles of technical change and efficiency change on these relationships, and the differences of these effects at the national and regional levels. Specifically, this study adopted a panel data of 30 Chinese provincial-level areas between 2000 and 2018 to investigate the impacts of environmental regulations on green productivity in China, with the consideration of the moderating effects of technical change and efficiency change. Additionally, we conducted this study in the context of China, a typical emerging economy, which has significant regional heterogeneous characteristic, and thus we also considered the regional heterogeneity in empirical analysis [11]. Last but not least, the temporal inertia usually leads to path dependence, which may increase the difficulties of optimizing the efficiency and increasing the total carbon productivity from the long run [12]. Thus, to avoid the potential biased estimation, the dynamic and spatial spillover effects should not be ignored in empirical analysis.

The potential marginal contributions of this study are threefold. Firstly, this study has comprehensively analyzed the impacts of heterogeneous environmental regulations on green productivity in China based on the Porter Hypothesis, which had been explored in previous studies, thus contributing to the literature of environmental economics. Secondly, this study has identified the moderating roles of technical change and efficiency change on the relationships between different environmental regulations and green productivity in China, whose contingent effects have been ignored by prior research, thus delimiting the boundaries where environmental regulations play their roles and providing a better understanding of the relationship between environmental regulations and green productivity. Thirdly, this study has employed the static, dynamic, spatial, and dynamic spatial methods comprehensively, and further investigate the regional heterogeneity, which helps to provide more specific policy implications.

The reminder of this study is organized as follows. The literature review is presented in Section 2. The research data and methods are shown in Section 3. The empirical analysis and results are displayed in Section 4. Section 5 concludes the research and addresses some policy implications.

## 2. Literature Review

### 2.1. Environmental Regulations

The existing literature usually divides environmental regulations into different types [13]. For example, a relatively acknowledged viewpoint conceived that the environmental regulations can be classified into three kinds including command–control, market incentive, and public participation [10]. Specifically, the command–control environmental regulation includes the laws and regulations that force polluters to reduce emissions and save resources, including emission permits and discharge standards [11]. The market-incentive environmental regulation means the measures to encourage polluters to cut emissions, which include emission taxes, emission trading, pollution charges, and so on [14]. The public-participation environmental regulation denotes the non-mandatory measures that encourage polluters to control pollution, including consultation and negotiation [11].

Currently, scholars have analyzed the effectiveness of environmental regulation from the perspective of carbon emissions. In terms of the effects of environmental regulations on carbon emissions, there coexists two contradictory views, including the green paradox effect and the reverse emission reduction effect. For example, Sinn (2008) argued that certain environmental regulations could aggravate the problem of global warming [15]. This view was consistent with the research of Smulders et al. (2012) [16] and Ritter and Schopf (2014) [17]. However, Van der Ploeg and Withagen (2012) revealed that environmental regulations could inhibit carbon emissions [18], which was also supported by Cairns (2014) [19] and Guo and Wang (2018) [20] However, Zhang et al. (2020) found that in the short term, environmental regulation can increase carbon emissions while in the long term, environmental regulation intensity could increase and enhance pollution control, thus reducing carbon emissions [1]. In addition, it should be noted that some studies also explored the influencing mechanisms of environmental regulation on carbon emissions from the perspectives of technology innovation [21,22], foreign direct investment [1], and industrial structure [2], which provides a copious reference for this study.

Many scholars have explored the effect of environmental regulations on green innovation. For example, Steinhorst and Matthies (2016) found that environmental regulations could positively affect green innovation through the promotion of public interest and investment [23], which was supported by the studies of Xu (2017) [24] and Giessen and Sahide [25]. However, other scholars hold the opposite view and contended that environmental regulations cannot enhance green innovation, but rather inhibited green innovation as a result of the increased innovation cost. For example, Stucki et al. (2018) found that environmental regulations would increase the investment into green innovation and thus affected it negatively [26].

### 2.2. Green Productivity

Technically, the calculation methods of total factor productivity (TFP) can be largely divided into two types: parametric and nonparametric. However, in most of the parametric research, based on the Solow residuals of Cobb–Douglas (CD) production function, TFP is merely calculated by utilizing capital and labor as inputs and GDP as the desirable output into the production function, not only neglecting the energy inputs and their undesirable impacts, but also limiting by the strict assumptions of complete market competition, Hicks-neutral technical progress, and constant returns to scale [12]. On the other hand, data envelopment analysis (DEA), a nonparametric technique that does not have the requirement for the specific assumptions on the distance function, has been widely used to estimate the value of TFP.

In particular, the directional distance function (DDF) method, which credits reductions in bad and increases in good in the estimation of a production frontier simultaneously under the framework of DEA, is employed to measure green productivity or other similar indexes at both macro and micro levels. For example, Lin et al. (2013) used a DDF model to measure the environmental productivity in 70 countries over the period from 1981 to 2007 with consideration for both desirable output (i.e., GDP) and undesirable output (i.e., greenhouse gas) [27]. Chen and Golley (2014) estimated the changing patterns of green total factor productivity growth of 38 industrial sectors in China during 1980 and 2010 by using the DDF method [7].

However, DDF has the disadvantage of ignoring the possible slacks, which could result in the overestimation of efficiency [28]. Accordingly, the slack-based measure (SBM) method has been put forward to measure green productivity, and these two kinds of methods were combined to calculate green productivity. For example, Liu and Xin (2019) used the SBM–DDF method to evaluate the provincial green productivity of 17 primary provinces in China [29]. Even so, the SBM–DDF model fails to effectively solve the problem of the inconsistency of production frontier in each production unit stage, thus influencing the comparability of inter-temporal results.

Therefore, some scholars have adopted the Malmquist–Luenberger (ML) index, which can overcome the above-mentioned limitations, to measure green productivity. For example, Li and Lin (2017) applied the ML index to reflect green productivity when investigating the impacts of investment-driven economic growth models based on the data from 30 provinces in China over the period from 1997 to 2010 [30]. Bampatsou and Halkos (2018) utilized both the traditional Malmquist–Luenberger and bootstrapped Malmquist productivity indexes to conduct the nonparametric frontier analysis based on the data of the 28 EU countries during 1993 and 2015 [31]. Zhang and Vinge (2021) adopted the ML method to measure green productivity and found the role of innovation efficiency in achieving sustainable development [6]. Hence, the ML index is employed to measure green productivity in this study.

### 2.3. Environmental Regulations and Green Productivity

Prior studies on the relationship between environmental regulations and green productivity have asserted two streamlines that included the non-linear and positive relationships. As for the non-linear relationship, Peng (2020) explored the relationships among environmental regulations, green technology innovation, green technology progress, and green productivity by using the data of 274 prefecture-level cities in China during 2005 and 2015 and found that the environmental regulation promoted local green productivity, while the weighted environmental regulation in the adjacent regions inhibited green productivity [9]. In addition, Xie et al. (2017) found that both command-and-control and market-based regulations had non-linear relationships with green productivity [5]. Li and Ramanathan (2018) also explored the effects of command-and-control regulations, market-based regulations and informal regulations on green productivity and revealed that only command-and-control and market-based regulations had non-linear and positive effects on green productivity [32].

With regard to the positive relationship, Guo et al. (2017) investigated the relationship between environmental regulations and regional green productivity growth and concluded that environmental regulations could not directly promote regional green productivity growth, while technology innovation driven by environmental regulations had a positive impact on regional green productivity [33]. In addition, Li and Wu (2017) investigated the influence of both local and civil environmental regulations on green productivity in China by using the panel data of 273 cities during the period of 2003–2013, and they found that civil environmental regulations had both direct and indirect positive effects on green productivity [8]. Moreover, Ai et al. (2020) explored the influence of environmental regulations on green productivity from the micro perspective and contended that environmental regulations can present a positive influence on enterprise green productivity [34].

Therefore, after the exploration of the literature, it can be found that there are few empirical studies exploring the effects of heterogeneous environmental regulations on green productivity, and most studies only examined the effects of one or two kinds of environmental regulations which leaves an opportunity to identify the influencing mechanism. Additionally, technical change and efficiency change may play significant roles in these relationships. In addition, the impact of environmental regulations on green productivity may be differentiated at the regional level, which are also worthy of investigation. In light of this, with the consideration of regional heterogeneity, the current study aims to investigate the heterogeneous effects of different kinds of environmental regulations on green productivity in China at national and regional levels and explore the moderating effects of technical change and efficiency change on these relationships simultaneously.

## 3. Data and Methods

### 3.1. Variables Calculation

#### 3.1.1. Dependent Variable

This study adopts the Malmquist–Luenberger (ML) index to measure green productivity by employing the MaxDEA 8.0 software (MaxDEA Software Ltd, Beijing, China). The input indicators contain labor (calculated by the total number of total employments at the end of the year), capital stock (calculated by the perpetual inventory method), and energy consumption (calculated by the number of primary energy consumption). The desirable output indicator is the real GDP based on 2000. The desirable output indicators are the total number of greenhouse gas emissions (i.e., CO_2_) and the annual provincial-level average concentration of fine particles with a diameter of 2.5 μm or less (i.e., PM2.5).

#### 3.1.2. Moderating Variables

The Malmquist–Luenberger (ML) index has been decomposed into two indicators, technical change (TC) and efficiency change (EC), in this study. To dig out the moderating effect of technical change (TC) and efficiency change (EC) on the nexus between environmental regulation and green productivity, the interactive terms of those two decomposition indicators and heterogeneous environmental regulations (i.e., command–control, market incentive, and public participation) are introduced into the equation, respectively. In addition, to eliminate the estimation bias caused by collinearity, all the interaction items are centralized.

#### 3.1.3. Key Explanatory Variables

Referring to the study of Feng and Cheng (2018) [10], three types of heterogeneous environmental regulations (i.e., command–control, market incentive, and public participation) are employed in this study, which are measured by the number of environmental punishment cases per 10^4^ people, sewage charge per capita, and the number of the National People’s Congress (NPC) suggestions and the Chinese People’s Political Consultative Conference (CPPCC) proposals per 10^4^ people, respectively.

#### 3.1.4. Control Variables

Except for the moderating and independent variables, several other factors that may have important effects on the dependent variable are included in the equations as the control variables such as population density (PD), economic development (ED), energy consumption structure (ECS), government intervention (GI), ownership structure (OS), infrastructure (INF), and foreign direct investment (FDI), which are measured by the share of total population to local area, per capita GDP, the share of fossil fuel consumption in the total energy, the share of fiscal expenditure to local GDP, the share of employees worked in the state-owned enterprises to the total employees, the length of per capita roads, and the share of actual foreign direct investment in the local GDP, respectively.

### 3.2. Data Source

A panel data including 30 provincial-level regions (including provinces, autonomous regions, and municipalities directly under the central government) scanning from 2000 to 2018 are employed in this study, while four regions including Tibet, Hong Kong, Macao, and Taiwan are excluded as a result of data unavailability. The statistical data are mainly collected from the China Energy Statistical Yearbook, the China Statistical Yearbook, the China Environmental Statistical Yearbook, and the statistical yearbooks of various provincial-level regions. In particular, the annual average concentration of fine particles with a diameter of 2.5 μm or less (i.e., PM2.5) is extracted from the website of the Atmospheric Composition Analysis Group (ACAG) at Dalhousie University (http://fizz.phys.dal.ca/~atmos/martin/?page_id=140) (accessed on 1 March 2021). In addition, the nominal price indexes such as the market-incentive environmental regulation (i.e., per capita sewage discharge) and economic development (i.e., per capita GDP) are deflated by the 2000 constant-price index. The descriptive statistics of all variables are reported in Table 1. 

### 3.3. Model Specification

To figure out the direct effect of environmental regulation on green productivity, the fixed OLS model is set as follows:(1)$$ML_{it}=\alpha_{1}ER_{it}+\alpha_{2}M_{it}+\beta X_{it}+c_{0}+\varepsilon_{it}$$

In the formula, *ML_it_* denotes the degree of green productivity, which is expressed by the Malmquist–Luenberger index. *ER_it_* denotes heterogeneous environmental regulations (i.e., command–control, market incentive, and public participation), *M_it_* denotes the moderating variables including technical change (*TC*) and efficiency change (*EC*), *X_it_* denotes a vector of control variables including population density (*PD*), economic development (*ED*), energy consumption structure (*ECS*), government intervention (*GI*), ownership structure (*OS*), infrastructure (*INF*), and foreign direct investment (*FDI*). *α*_1_, *α*_2_, and *β* denote the corresponding coefficients. *c*_0_ denotes the constant term. *ε_it_* denotes the disturbance term of the model.

In addition, to study the moderating roles of technical change and efficiency change, another common nonspatial model including the interaction term of environmental regulation and technical change or efficiency change is given as follows:(2)$$ML_{it}=\alpha_{1}ER_{it}+\alpha_{2}M_{it}+\alpha_{3}M_{it}*ER_{it}+\beta X_{it}+c_{0}+\varepsilon_{it}$$

In the formula, *M_it_***ER_it_* is the interaction term of the moderating variable and the independent variable, which aims to test the degree of the moderating effects of technical change (*TC*) and efficiency change (*EC*) on the nexus between environmental regulation and green productivity. In addition, in order to avoid the problem of multicollinearity caused by the interaction term, when introducing it, this study subtracts their average values from the moderating variable and the independent variable, respectively. *α*_3_ is the coefficient of *M_it_***ER_it_*. The other parameters are consistent with Equation (1).

Additionally, considering the possible path dependence of green productivity and referring to the research of Wu et al. (2019) on the dynamic econometric model [35], the one-phase lag term of green productivity is included in the two common nonspatial models above. However, the *p*-values of the AR(2) test are all lower than 0.1, indicating that the second-order serial correlation should not be ignored. Fortunately, after adding both the one-phase lag term and the two-phase lag term of green productivity to the model, the second-order serial correlation has no longer been supported. Therefore, the following two dynamic panel models are constructed: (3)$$ML_{it}=\tau_{1}ML_{i,t-1}+\tau_{2}ML_{i,t-2}+\alpha_{1}ER_{it}+\alpha_{2}M_{it}+\beta X_{it}+c_{0}+\varepsilon_{it}\mathrm{Shehzad}\;$$
(4)$$ML_{it}=\tau_{1}ML_{i,t-1}+\tau_{2}ML_{i,t-2}+\alpha_{1}ER_{it}+\alpha_{2}M_{it}+\alpha_{3}M_{it}*ER_{it}+\beta X_{it}+c_{0}+\varepsilon_{it}$$

In the formula, *ML_i,t−_*_1_ and *ML_i,t−_*_2_ denote lag terms of green productivity, which can not only be considered as proxy variables for some omitted variables but also reflect the inertia effect of *ML_it_*. *τ*_1_ and *τ*_2_ denote the corresponding coefficients of *ML_i,t−_*_1_ and *ML_i,t−_*_2_, respectively. The other parameters are consistent with Equation (2). As for the estimation methods of the dynamic non-spatial panel model, the two-step SYS-GMM method and the two-step DIFF-GMM method are commonly used in the existing literature. Compared with the two-step DIFF-GMM method, the two-step SYS-GMM method estimates the coefficients of the model that do not change with time, which improves the estimation efficiency and makes the results more accurate. Hence, this study has employed the two-step SYS-GMM method to conduct the dynamic non-spatial econometric model.

Moreover, considering the possibility of spatial spillover effects, this study further modifies the two basic formulas, and the spatial Durbin model is employed here, as shown in Equation (5) and Equation (6) on the basis of Equation (1) and Equation (2).
(5)$$\begin{array}{l} ML_{it}=\rho W*ML_{it}+\alpha_{1}ER_{it}+\alpha_{2}M_{it}+\beta X_{it}+\theta_{1}W*ER_{it}\\ +\theta_{2}W*M_{it}+\varphi W*X_{it}+\varepsilon_{it} \end{array}$$
(6)$$\begin{array}{l} ML_{it}=\rho W*ML_{it}+\alpha_{1}ER_{it}+\alpha_{2}M_{it}+\alpha_{3}M_{it}*ER_{it}+\beta X_{it}\\ +\theta_{1}W*ER_{it}+\theta_{2}W*M_{it}+\theta_{3}W*ER_{it}*M_{it}+\varphi W*X_{it}+\varepsilon_{it} \end{array}$$

In the formula, *W* denotes the spatial weight matrix. *ρ* denotes the spatial auto-regressive coefficient of the dependent variable that is environmental regulation. *θ*_1_, *θ*_2_, and *θ*_3_ denote the corresponding spatial coefficients of the independent variable, th emoderating variable, and the interaction term between the independent variable and moderating variable. *φ* denotes the spatial coefficients of the control variables. The other parameters are also consistent with Equation (2).

Furthermore, considering the possibility of both temporal dynamic effects and spatial spillover effects, this study also adopts the dynamic spatial Durbin model, and the respective models are modified as follows:(7)$$\begin{array}{l} ML_{it}=\tau_{1}ML_{i,t-1}+\rho W*ML_{it}+\alpha_{1}ER_{it}+\alpha_{2}M_{it}+\beta X_{it}\\ +\theta_{1}W*ER_{it}+\theta_{2}W*M_{it}+\varphi W*X_{it}+\varepsilon_{it} \end{array}$$
(8)$$\begin{array}{l} ML_{it}=\tau_{1}ML_{i,t-1}+\rho W*ML_{it}+\alpha_{1}ER_{it}+\alpha_{2}M_{it}+\alpha_{3}M_{it}*ER_{it}+\beta X_{it}\\ +\theta_{1}W*ER_{it}+\theta_{2}W*M_{it}+\theta_{3}W*ER_{it}*M_{it}+\varphi W*X_{it}+\varepsilon_{it} \end{array}$$

Last but not least, it is important and necessary to note that the spatial weight matrix reflects the specific spatial correlation form between different spatial spaces, which has been acknowledged as a core element for the spatial econometric model. In general, the spatial weight matrices (i.e., the first-order adjacency weight matrix and the geographical inverse distance weight matrix) were set in symmetric. However, the provinces with a higher level of economic development undoubtedly have a stronger influence on the other provinces with lower levels of economic development, thus the spatial economic correlation is not completely symmetrical. In addition, due to the existence of island provinces (i.e., Hainan), the adjacency weight matrix could not completely capture the spatial correlation. Therefore, referring to the research of [36], this study has constructed an asymmetric geographical economic weight matrix with the consideration of both geographical and economic spatial correlations, which is defined as follows:(9)$$W=\{\begin{array}{ll} \frac{1}{d_{ij}^{2}}\frac{PGDP_{j}}{PGDP_{i}}, & if\;i\neq j\\ 0, & if\;i=j \end{array}$$
where *d_ij_* denotes the geo-centroid distance of province *i* and province *j*, and the squared inverse distance implies that the spatial correlation between regions will decrease sharply as geographical distance increases. *PGDP_i_* denotes the average per capita GDP of province *i* during the research period.

## 4. Empirical Results and Analysis

### 4.1. Full Sample Estimation

For comparison, the fixed OLS model, the dynamic SYS-GMM model, the static spatial Durbin model, and the dynamic spatial Durbin model are used for parameter estimation. According to the empirical results in Table 2, Table 3, Table 4, Table 5, Table 6 and Table 7, it is shown that green productivity has significant temporal lag effects in the estimation results of the dynamic SYS-GMM model and spatial spillover effects in the estimation results of the static spatial Durbin model, respectively. Thus, the temporal lag effects and the spatial spillover effects of green productivity should not be ignored in analysis; that is, the results of the dynamic spatial Durbin model, which considers the dynamic and spatial spillover effects of green productivity simultaneously, have the best theoretical expectations and measurement techniques. Hence, this study will focus on the estimation results of the dynamic spatial Durbin model in the following discussion.

First, all the temporal lag coefficients of green productivity are insignificant, while all the spatial lag coefficients of green productivity are significantly positive; in other words, considering the spatial spillover effects and the temporal lag effects of green productivity simultaneously, the negative path-dependent feature is not supported any longer, while the spatial spillover effect is still the power source for promoting green productivity in China. In addition, all the direct coefficients of the moderating variables are significantly positive, while the absolute values of technical change are greater than the corresponding values of efficiency change; in other words, compared with efficiency change, technical change has a stronger promotion effect on green productivity in China.

Second, the indirect coefficient of the interactive term is significantly negative in Table 3, while the direct coefficients of the key explanatory variable and the corresponding interactive term in Table 2 and Table 3 and the indirect coefficient of the interactive term in Table 2 are insignificant; in other words, except for the negative moderation effect of efficiency change on the nexus between command–control environmental regulation and green productivity in surrounding provinces, not only the direct and indirect effects of command–control environmental regulation on green productivity in China but also the moderating role of technical change are relatively weak.

Third, the direct coefficients of the independent variable are insignificant in Table 4 and Table 5, and the indirect coefficients of the independent variable are insignificant in Table 4 and significantly positive in Table 5; in other words, the direct effect of market-incentive environmental regulation on green productivity in local provinces is relatively weak, while the corresponding indirect effect is unstable. In addition, the direct coefficient of the interactive term is positive and significant in Table 4 and insignificant in Table 5, while the direct coefficient of the interactive term is negative and significant in Table 4 and insignificant in Table 5; in other words, with the consideration of the moderating variable, technical change positively moderates the nexus between market-incentive environmental regulation and green productivity in local provinces and negatively moderates this nexus in surrounding provinces, while the moderating role of efficiency change is not supported.

Fourth, all the direct coefficients of the independent variable are insignificant in Table 6 and Table 7, and all the indirect coefficients of the independent variable are negative in Table 6 and Table 7, while only the indirect coefficient of the key explanatory variable in the last column of Table 7 is significant at the 10% level; in other words, the direct effect of public-participation environmental regulation on green productivity in local provinces is relatively weak, while the corresponding indirect effect to some extent inhibits green productivity in surrounding provinces. In addition, the direct coefficient of the interactive term is insignificant in Table 6 and significantly positive in Table 7, while the indirect coefficient of the interactive term is significantly positive in Table 6 and insignificant in Table 7; in other words, with the consideration of the moderating variable, technical change positively moderates the nexus between public-participation environmental regulation and green productivity in surrounding provinces, while efficiency change positively moderates this nexus in local provinces.

### 4.2. Subsample Estimations in Different Regions

To investigate whether the regional heterogeneity is supported or not, we divide the total sample into two groups, the eastern region group and the central and western regions group, and we employ the dynamic spatial Durbin model for estimation and analysis here.

The moderating results of technical change in the eastern region are shown in Table 8. First, all the temporal lag and spatial lag coefficients of green productivity are positive but insignificant; in other words, when technical change is employed as the moderating variable, the spatial spillover effect of green productivity is suppressed; that is, technical change has a stronger explanation in the eastern region. Second, the direct coefficients of environmental regulation in columns (3) and (4) are negative and significant, and the indirect coefficient of it in column (1) is significantly positive, while all the other direct and indirect coefficients of it are insignificant; in other words, when technical change is employed as the moderating variable, only market-incentive environmental regulation has a direct and negative impact on green productivity. Third, the direct coefficient of the interactive term is significantly positive in column (2), and the indirect coefficient of it is significantly negative in column (2), while all the other direct coefficients of it are insignificant in columns (4) and (6); in other words, with the consideration of the moderating variable, technical change only positively moderates the nexus between command–control environmental regulation and green productivity in local provinces and negatively moderates this nexus in surrounding provinces.

The moderating results of efficiency change in the eastern region are shown in Table 9. First, all the spatial lag coefficients of green productivity are significantly positive, and all the temporal lag coefficients of it are negative, while only the temporal lag coefficients of it in columns (5) and (6) are significant; in other words, when efficiency change is employed as the moderating variable, the negative path-dependence and the spatial spillover effect of green productivity are also supported in the eastern region. Second, all the direct coefficients of environmental regulation are insignificant, and all the indirect coefficients of it are negative, while only the indirect coefficients of it in columns (3) and (4) are significant; in other words, when efficiency change is employed as the moderating variable, only market-incentive environmental regulation has an indirect and negative impact on green productivity. Third, the direct coefficients of the interactive term are negative and significant in column (2) and insignificant in columns (4) and (6), and the indirect coefficients of it are negative and significant in columns (2) and (4) and insignificant in column (6); in other words, with consideration of the moderating variable, efficiency change not only negatively moderates the nexus between command–control environmental regulation and green productivity in local and surrounding provinces, but also negatively moderates the nexus between market-incentive environmental regulation and green productivity in surrounding provinces. 

The moderating results of technical change in the central and western regions are shown in Table 10. First, all the temporal lag coefficients of green productivity are insignificant, while all the spatial lag coefficients of it are positive and significant; in other words, when technical change is employed as the moderating variable, only the spatial spillover effect of green productivity is supported in the central and western regions. Second, all the direct and indirect coefficients of environmental regulation are insignificant; in other words, when technical change is employed as the moderating variable, both the direct and indirect effects of environmental regulations on green productivity are relatively weak. Third, the direct coefficient of the interactive term is positive and significant in column (4), and the indirect coefficients of it is negative and significant in column (4) and (6), while the other direct and indirect coefficients of it are insignificant; in other words, with the consideration of the moderating variable, technical change not only positively moderates the nexus between market-incentive environmental regulation and green productivity in local provinces and negatively moderates this nexus in surrounding provinces, but also positively moderates the nexus between public-participation environmental regulation and green productivity in surrounding provinces.

The moderating results of efficiency change in the central and western regions are shown in Table 11. First, all the spatial lag coefficients of green productivity are significantly positive, and all the temporal lag coefficients of it are negative, while only the temporal lag coefficients of it in columns (3) and (4) are significant; in other words, when efficiency change is employed as the moderating variable, the negative path-dependence and the spatial spillover effect of green productivity are also supported in the central and western regions. Second, all the direct coefficients of environmental regulation are insignificant, and the indirect coefficients of it in columns (3) and (4) are positive and significant, while the other indirect coefficients of it in columns (1), (2), (5), and (6) are negative and insignificant; in other words, when efficiency change is employed as the moderating variable, only market-incentive environmental regulation has an indirect and positive impact on green productivity. Third, all the direct and indirect coefficients of the interactive term are insignificant; in other words, with the consideration of the moderating variable, efficiency change fails to moderate the nexus between environmental regulations and green productivity in local and surrounding provinces. 

## 5. Conclusions, Policy Implications, and Research Prospects

### 5.1. Conclusions

This study investigates the effects of heterogeneous environmental regulations on green productivity and the moderating roles of technical change and efficiency change by employing the dynamic spatial Durbin model based on the panel data of 30 provinces in China from 2000 to 2018. According to the above discussed empirical results, the conclusions of this study can be summarized as follows:

For the whole sample, compared with efficiency change, technical change has a stronger promotion effect on green productivity in China. In addition, considering the spatial spillover effects and the temporal lag effects of green productivity simultaneously, the negative path-dependent feature is not supported any longer, while the spatial spillover effect is still the power source for promoting green productivity in China. This result further extends the study of Peng (2020) that the environmental regulation in the adjacent regions could inhibit green productivity [9]. Moreover, the moderating roles of technical change and efficiency change for the nexus between heterogeneous environmental regulations and green productivity in China are partly and conditionally supported.

For the subsample, compared with efficiency change, technical change has a stronger moderating effect on the nexus between heterogeneous environmental regulations and green productivity at the regional level. This result is consistent with the research of Guo et al. (2017), who figured out that technology innovation resulting from environmental regulations positively affects green productivity [33]. In addition, the effects of heterogeneous environmental regulations on green productivity at the regional level have the feature of spatial heterogeneity. In particular, market-incentive environmental regulation has lost momentum in promoting green productivity in the surrounding provinces of the eastern region, while market-incentive environmental regulation has effectively promoted green productivity in the surrounding provinces of the central and western regions. This result further extends the research of Li and Wu (2017), which mainly revealed the influence of environmental regulations on green productivity at the national level [8].

### 5.2. Policy Implications

The above-mentioned findings imply and inspire the policy implications as follows: 

First, considering the heterogeneous effects of different kinds of environmental regulations on green productivity in China, environmental regulations should be specially designed across different regions to utilize their special advantages. Specifically, for the regions where command–control environmental regulations could strengthen green productivity, the local governments could issue rules and establish standards for the discharge and disposal of pollutants and levy taxes on environmental pollution. For the regions where market-incentive environmental regulations could improve green productivity, the local governments could promote the development of emission trading systems and thus weaken the externality of economic development to the environment. For the regions where public-participation environmental regulations could accelerate green productivity, the local governments could promote the propaganda on environmental protection and encourage firms to voluntarily engage in the sustainable development of the environment.

Second, to enjoy the bonus of spatial spillover effect and weaken the negative path dependence of green productivity in China, special emphasis should be placed on how to achieve green development by taking the spatial attributes of resource endowments and industrial agglomeration into consideration. Specifically, the governments could guide the effective transfer of resources and lower local resource redundancies considering the location advantage of local resource endowments and plan strategically the local industrial layouts to generate a high-quality industrial agglomeration, thus boosting its positive spatial spillover effect to green development. For the eastern regions, they should transfer the advanced green manufacturing and management technologies to the middle and western regions and make full use of the market-based environmental regulations as for their developed market and legal systems. For the middle and western regions, they should establish green development strategies and incorporate the green development performance into the government performance appraisal systems.

Third, to deal with the differentiated moderating effects of technical change and efficiency change on the nexus between heterogeneous environmental regulations and green productivity in China, the establishment and selection of environmental regulations should consider the diverse effects of technical change and efficiency change. Specifically, to stimulate the effectiveness of technical change in the relationship between environmental regulations and green productivity, the design of environmental regulations should be combined with the levels of local technology progress and technical change. The local governments should encourage and guide the research and development of green technologies and introduce private capitals into it to boost the green technological innovation. To release and improve the potential of efficiency change in promoting green productivity in China, the implementation of environmental regulations should be coordinated with the upgrading of industrial structure and the elimination of backward production capacity. The governments should restrict and eliminate the industries with backward production capability and upgrade the traditional industries into the two ends of the value chain by accelerating production efficiency.

### 5.3. Research Prospects

This study should also be interpreted with some limitations, which in turn gives some recommendations for future research. First, due to data restrictions, provincial panel data is employed in this study. To confirm our findings, the prefecture-level and county-level panel data can be used for a comprehensive and thorough analysis. Second, in our study, environmental regulations are divided into three types based on the difference of regulated executors, and each type of environmental regulations is measured by a single indicator in empirical research. In further research, an expansion of the compound indicator system may be considered to obtain more guiding conclusions. Third, according to the calculation method of [28], the unfeasible solution of the ML index can be merely solved with constant returns to scale, but not with variable returns to scale. Thus, a more advanced calculation method of efficiency should be employed in the future. Last but not least, the spatial Durbin model is adopted to do the empirical analysis in this paper, but individual effect is ignored. To expand the research, the Geographically and Temporally Weighted Regression (GTWR) model could be adopted to empirically study the impact of environmental regulations on green productivity in China and other developing countries undergoing similar urbanization and industrialization processes.

## Figures and Tables

**Table 1 ijerph-18-11449-t001:** Descriptive statistics.

Variables	Unit	N	Mean	S.D.	Min	Max
ML	—	540	1.064	0.141	0.504	1.939
TC	—	540	1.049	0.125	0.543	1.939
EC	—	540	1.015	0.079	0.717	1.658
ER_Command–control	10^4^ Person/piece	540	1.979	2.257	0.015	16.636
ER_Market incentive	Yuan	540	12.577	10.043	1.457	84.687
ER_Public participation	10^4^ Person/piece	540	0.107	0.066	0.004	0.724
PD	10^3^ Person/km^2^	540	2.388	1.357	0.056	6.307
ED	10^4^ Yuan	540	3.331	2.591	0.300	15.310
ECS	—	540	0.461	0.158	0.016	0.804
GI	—	540	0.211	0.104	0.077	0.758
OS	—	540	0.586	0.126	0.192	0.901
INF	Meter	540	2.986	2.203	0.000	13.991
FDI	—	540	0.024	0.020	0.000	0.146

**Table 2 ijerph-18-11449-t002:** The moderating results of technical change for the nexus between command–control environmental regulation and green productivity.

Variables	OLS Model (FE)	Dynamic Non-Spatial Model (SYS-GMM)	Static Spatial Durbin Model	Dynamic Spatial Durbin Model	OLS Model (FE)	Dynamic Non-Spatial Model (SYS-GMM)	Static Spatial Durbin Model	Dynamic Spatial Durbin Model
	(1)	(2)	(3)	(4)	(5)	(6)	(7)	(8)
*ML_i,t−_* _1_		−0.041 *		0.014		−0.024		0.018
		(−1.696)		(0.595)		(−0.935)		(0.734)
*ML_i,t−_* _2_		−0.105 ***				−0.095 ***		
		(−4.899)				(−3.627)		
*W*ML_it_*			0.137 ***	0.138 ***			0.136 ***	0.137 ***
			(3.117)	(3.052)			(3.097)	(3.030)
*ER_it_*	0.002	0.000	0.002	0.002	0.001	0.001	0.001	0.001
	(0.835)	(0.429)	(0.770)	(0.760)	(0.522)	(0.821)	(0.539)	(0.558)
*M_it_*	0.930 ***	0.924 ***	0.856 ***	0.830 ***	0.926 ***	0.937 ***	0.850 ***	0.827 ***
	(33.964)	(58.694)	(28.192)	(24.450)	(33.324)	(33.414)	(27.403)	(24.181)
*M_it_*ER_it_*					−0.014	−0.006	−0.014	−0.011
					(−1.072)	(−0.654)	(−1.058)	(−0.827)
*W*ER_it_*			−0.003	−0.004			−0.003	−0.004
			(−0.954)	(−1.236)			(−0.933)	(−1.260)
*W*M_it_*			0.028	0.065			0.032	0.064
			(0.512)	(1.126)			(0.561)	(1.091)
*W*M_it_*ER_it_*							0.001	−0.006
							(0.034)	(−0.256)
Constant	0.172 ***	0.388 ***			1.142 ***	1.481 ***		
	(2.627)	(3.021)			(19.136)	(11.012)		
Control variables	Yes	Yes	Yes	Yes	Yes	Yes	Yes	Yes
AR(1)		[0.000]				[0.000]		
AR(2)		[0.296]				[0.320]		
Sargan		[1.000]				[1.000]		
R-squared	0.709		0.636	0.597	0.710		0.644	0.608
N	540	480	540	510	540	480	540	510

Note: (1) Robust t-statistics in the parentheses of columns (1), (2), (5), and (6); z-statistics in the parentheses of columns (3), (4), (7), and (8); (2) *** and * denote *p* < 0.01 and *p* < 0.1, respectively; (3) the *p*-values are shown in square brackets.

**Table 3 ijerph-18-11449-t003:** The moderating results of efficiency change for the nexus between command–control environmental regulation and green productivity.

Variables	OLS Model (FE)	Dynamic Non-Spatial Model (SYS-GMM)	Static Spatial Durbin Model	Dynamic Spatial Durbin Model	OLS Model (FE)	Dynamic Non-Spatial Model (SYS-GMM)	Static Spatial Durbin Model	Dynamic Spatial Durbin Model
	(1)	(2)	(3)	(4)	(5)	(6)	(7)	(8)
*ML_i,t−_* _1_		−0.201 ***		−0.023		−0.198 ***		−0.020
		(−5.063)		(−0.736)		(−4.514)		(−0.622)
*ML_i,t−_* _2_		−0.288 ***				−0.338 ***		
		(−7.836)				(−6.149)		
*W*ML_it_*			0.432 ***	0.462 ***			0.431 ***	0.460 ***
			(12.727)	(13.635)			(12.669)	(13.568)
*ER_it_*	−0.006	−0.004 **	−0.005	−0.004	−0.006 *	−0.005 ***	−0.004	−0.003
	(−1.621)	(−2.441)	(−1.289)	(−1.157)	(−1.736)	(−3.149)	(−1.176)	(−1.023)
*M_it_*	0.767 ***	0.666 ***	0.706 ***	0.709 ***	0.757 ***	0.768 ***	0.697 ***	0.699 ***
	(10.711)	(4.687)	(12.045)	(12.746)	(10.555)	(5.804)	(11.815)	(12.520)
*M_it_*ER_it_*					−0.051	0.035	−0.009	−0.006
					(−1.460)	(1.180)	(−0.326)	(−0.239)
*W*ER_it_*			0.001	0.001			0.001	0.000
			(0.271)	(0.141)			(0.182)	(0.025)
*W*M_it_*			−0.060	−0.068			−0.100	−0.107
			(−0.711)	(−0.851)			(−1.132)	(−1.282)
*W*M_it_*ER_it_*							−0.063	−0.066 *
							(−1.463)	(−1.646)
Constant	0.301 **	0.577 *			1.071 ***	1.078 ***		
	(2.418)	(1.892)			(11.073)	(5.033)		
Control variables	Yes	Yes	Yes	Yes	Yes	Yes	Yes	Yes
AR(1)		[0.001]				[0.001]		
AR(2)		[0.168]				[0.076]		
Sargan		[1.000]				[1.000]		
R-squared	0.218		0.134	0.139	0.222		0.130	0.129
N	540	480	540	510	540	480	540	510

Note: (1) Robust t-statistics in the parentheses of columns (1), (2), (5), and (6); z-statistics in the parentheses of columns (3), (4), (7), and (8); (2) ***, **, and * denote *p* < 0.01, *p* < 0.05, and *p* < 0.1, respectively; (3) the *p*-values are shown in square brackets.

**Table 4 ijerph-18-11449-t004:** The moderating results of technical change for the nexus between market-incentive environmental regulation and green productivity.

Variables	OLS Model (FE)	Dynamic Non-Spatial Model (SYS-GMM)	Static Spatial Durbin Model	Dynamic Spatial Durbin Model	OLS Model (FE)	Dynamic Non-Spatial Model (SYS-GMM)	Static Spatial Durbin Model	Dynamic Spatial Durbin Model
	(1)	(2)	(3)	(4)	(5)	(6)	(7)	(8)
*ML_i,t−_* _1_		−0.022 **		0.019		−0.016		0.017
		(−2.019)		(0.756)		(−0.658)		(0.690)
*ML_i,t−_* _2_		−0.088 ***				−0.076 ***		
		(−6.397)				(−3.420)		
*W*ML_it_*			0.135 ***	0.133 ***			0.137 ***	0.138 ***
			(3.054)	(2.936)			(3.116)	(3.051)
*ER_it_*	−0.000	0.001 ***	−0.000	−0.000	−0.000	0.000	−0.000	−0.001
	(−0.235)	(3.568)	(−0.251)	(−0.770)	(−0.406)	(0.496)	(−0.337)	(−0.849)
*M_it_*	0.930 ***	0.905 ***	0.857 ***	0.832 ***	0.932 ***	0.903 ***	0.873 ***	0.845 ***
	(33.526)	(44.738)	(28.118)	(24.278)	(33.604)	(26.029)	(27.957)	(24.559)
*M_it_*ER_it_*					0.004	0.006 ***	0.004	0.005 *
					(1.534)	(7.631)	(1.498)	(1.714)
*W*ER_it_*			−0.001	−0.000			−0.000	−0.000
			(−0.709)	(−0.313)			(−0.555)	(−0.037)
*W*M_it_*			0.037	0.074			0.022	0.062
			(0.679)	(1.277)			(0.394)	(1.071)
*W*M_it_*ER_it_*							−0.006**	−0.008 **
							(−2.081)	(−2.367)
Constant	0.174 ***	0.253 ***			1.144 ***	1.171 ***		
	(2.645)	(4.193)			(19.195)	(10.108)		
Control variables	Yes	Yes	Yes	Yes	Yes	Yes	Yes	Yes
AR(1)		[0.001]				[0.001]		
AR(2)		[0.322]				[0.251]		
Sargan		[1.000]				[1.000]		
R-squared	0.709		0.636	0.603	0.710		0.638	0.607
N	540	480	540	510	540	480	540	510

Note: (1) Robust t-statistics in the parentheses of columns (1), (2), (5), and (6); z-statistics in the parentheses of columns (3), (4), (7), and (8); (2) ***, **, and * denote *p* < 0.01, *p* < 0.05, and *p* < 0.1, respectively; (3) the *p*-values are shown in square brackets.

**Table 5 ijerph-18-11449-t005:** The moderating results of efficiency change for the nexus between market-incentive environmental regulation and green productivity.

Variables	OLS Model (FE)	Dynamic Non-Spatial Model (SYS-GMM)	Static Spatial Durbin Model	Dynamic Spatial Durbin Model	OLS Model (FE)	Dynamic Non-Spatial Model (SYS-GMM)	Static Spatial Durbin Model	Dynamic Spatial Durbin Model
	(1)	(2)	(3)	(4)	(5)	(6)	(7)	(8)
*ML_i,t−_* _1_		−0.139 ***		−0.052		−0.130 ***		−0.050
		(−6.327)		(−1.641)		(−3.388)		(−1.576)
*ML_i,t−_* _2_		−0.232 ***				−0.180 ***		
		(−14.386)				(−4.673)		
*W*ML_it_*			0.414 ***	0.450 ***			0.414 ***	0.449 ***
			(11.969)	(13.208)			(11.982)	(13.203)
*ER_it_*	0.003 ***	0.003 ***	0.001 *	0.001	0.003 ***	0.002 ***	0.001 *	0.001
	(3.643)	(7.204)	(1.815)	(1.461)	(3.397)	(6.212)	(1.665)	(1.186)
*M_it_*	0.766 ***	0.807 ***	0.711 ***	0.717 ***	0.767 ***	0.838 ***	0.710 ***	0.715 ***
	(10.832)	(10.893)	(12.194)	(13.049)	(10.830)	(7.701)	(12.174)	(13.024)
*M_it_*ER_it_*					0.003	0.005 ***	0.003	0.003
					(0.555)	(2.881)	(0.700)	(0.840)
*W*ER_it_*			0.002 **	0.003 ***			0.002 *	0.003 ***
			(2.158)	(2.998)			(1.728)	(2.623)
*W*M_it_*			−0.030	−0.037			−0.036	−0.041
			(−0.353)	(−0.464)			(−0.427)	(−0.510)
*W*M_it_*ER_it_*							0.004	0.002
							(0.765)	(0.301)
Constant	0.306 **	0.046			1.120 ***	1.198 ***		
	(2.483)	(0.160)			(11.514)	(11.593)		
Control variables	Yes	Yes	Yes	Yes	Yes	Yes	Yes	Yes
AR(1)		[0.001]				[0.001]		
AR(2)		[0.489]				[0.710]		
Sargan		[1.000]				[1.000]		
R-squared	0.234		0.161	0.180	0.235		0.162	0.183
N	540	480	540	510	540	480	540	510

Note: (1) Robust t-statistics in the parentheses of columns (1), (2), (5), and (6); z-statistics in the parentheses of columns (3), (4), (7), and (8); (2) ***, **, and * denote *p* < 0.01, *p* < 0.05, and *p* < 0.1, respectively; (3) the *p*-values are shown in square brackets.

**Table 6 ijerph-18-11449-t006:** The moderating results of technical change for the nexus between public-participation environmental regulation and green productivity.

Variables	OLS Model (FE)	Dynamic Non-Spatial Model (SYS-GMM)	Static Spatial Durbin Model	Dynamic Spatial Durbin Model	OLS Model (FE)	Dynamic Non-Spatial Model (SYS-GMM)	Static Spatial Durbin Model	Dynamic Spatial Durbin Model
	(1)	(2)	(3)	(4)	(5)	(6)	(7)	(8)
*ML_i,t−_* _1_		−0.066 ***		0.012		−0.067 ***		0.009
		(−4.222)		(0.496)		(−3.888)		(0.356)
*ML_i,t−_* _2_		−0.124 ***				−0.126 ***		
		(−5.620)				(−5.715)		
*W*ML_it_*			0.138 ***	0.138 ***			0.141 ***	0.142 ***
			(3.132)	(3.055)			(3.204)	(3.146)
*ER_it_*	0.009	−0.100 **	0.065	0.057	0.007	−0.089 **	0.060	0.053
	(0.140)	(−2.422)	(1.051)	(0.897)	(0.104)	(−2.016)	(0.971)	(0.848)
*M_it_*	0.929 ***	0.879 ***	0.854 ***	0.828 ***	0.925 ***	0.889 ***	0.837 ***	0.802 ***
	(33.959)	(35.511)	(28.162)	(24.397)	(30.675)	(26.645)	(25.170)	(21.297)
*M_it_*ER_it_*					−0.174	0.451	−0.652	−0.787
					(−0.304)	(0.561)	(−1.193)	(−1.339)
*W*ER_it_*			−0.028	−0.037			−0.030	−0.051
			(−0.264)	(−0.341)			(−0.282)	(−0.478)
*W*M_it_*			0.031	0.067			0.064	0.136 **
			(0.560)	(1.174)			(1.089)	(2.132)
*W*M_it_*ER_it_*							1.198	2.104 **
							(1.514)	(2.386)
Constant	0.174 ***	0.331 ***			1.151 ***	1.247 ***		
	(2.649)	(3.766)			(19.473)	(13.066)		
Control variables	Yes	Yes	Yes	Yes	Yes	Yes	Yes	Yes
AR(1)		[0.001]				[0.001]		
AR(2)		[0.389]				[0.406]		
Sargan		[1.000]				[1.000]		
R-squared	0.709		0.637	0.606	0.709		0.637	0.604
N	540	480	540	510	540	480	540	510

Note: (1) Robust t-statistics in the parentheses of columns (1), (2), (5), and (6); z-statistics in the parentheses of columns (3), (4), (7), and (8); (2) *** and ** denote *p* < 0.01 and *p* < 0.05 respectively; (3) the *p*-values are shown in square brackets.

**Table 7 ijerph-18-11449-t007:** The moderating results of efficiency change for the nexus between public-participation environmental regulation and green productivity.

Variables	OLS Model (FE)	Dynamic Non-Spatial Model (SYS-GMM)	Static Spatial Durbin Model	Dynamic Spatial Durbin Model	OLS Model (FE)	Dynamic Non-Spatial Model (SYS-GMM)	Static Spatial Durbin Model	Dynamic Spatial Durbin Model
	(1)	(2)	(3)	(4)	(5)	(6)	(7)	(8)
*ML_i,t−_* _1_		−0.138 ***		−0.029		−0.152 ***		−0.036
		(−4.215)		(−0.926)		(−4.513)		(−1.148)
*ML_i,t−_* _2_		−0.204 ***				−0.207 ***		
		(−7.009)				(−7.729)		
*W*ML_it_*			0.430 ***	0.460 ***			0.432 ***	0.465 ***
			(12.618)	(13.546)			(12.710)	(13.783)
*ER_it_*	−0.077	−0.195 ***	0.002	0.012	−0.053	−0.188 *	0.032	0.055
	(−0.731)	(−4.405)	(0.025)	(0.143)	(−0.498)	(−1.699)	(0.360)	(0.654)
*M_it_*	0.761 ***	0.690 ***	0.703 ***	0.706 ***	0.760 ***	0.697 ***	0.700 ***	0.702 ***
	(10.624)	(12.268)	(11.997)	(12.717)	(10.614)	(12.885)	(11.979)	(12.749)
*M_it_*ER_it_*					1.137	0.650	1.292 *	1.669 **
					(1.177)	(0.366)	(1.664)	(2.289)
*W*ER_it_*			−0.226	−0.185			−0.252	−0.248 *
			(−1.520)	(−1.319)			(−1.615)	(−1.672)
*W*M_it_*			−0.061	−0.071			−0.064	−0.090
			(−0.724)	(−0.887)			(−0.727)	(−1.074)
*W*M_it_*ER_it_*							−0.567	−1.516
							(−0.373)	(−1.029)
Constant	0.310 **	0.425 ***			1.070 ***	0.986 ***		
	(2.478)	(3.558)			(11.030)	(6.139)		
Control variables	Yes	Yes	Yes	Yes	Yes	Yes	Yes	Yes
AR(1)		[0.000]				[0.001]		
AR(2)		[0.875]				[0.759]		
Sargan		[1.000]				[1.000]		
R-squared	0.215		0.135	0.142	0.217		0.136	0.144
N	540	480	540	510	540	480	540	510

Note: (1) Robust t-statistics in the parentheses of columns (1), (2), (5), and (6); z-statistics in the parentheses of columns (3), (4), (7), and (8); (2) ***, **, and * denote *p* < 0.01, *p* < 0.05, and *p* < 0.1, respectively; (3) the *p*-values are shown in square brackets.

**Table 8 ijerph-18-11449-t008:** The moderating results of technical change in the eastern region.

Variables	Command–Control		Market-Incentive		Public-Participation	
	(1)	(2)	(3)	(4)	(5)	(6)
*ML_i,t−_* _1_	0.013	0.012	0.017	0.023	0.009	0.007
	(0.375)	(0.378)	(0.480)	(0.645)	(0.247)	(0.201)
*W*ML_it_*	0.032	0.062	0.030	0.031	0.002	0.003
	(0.386)	(0.753)	(0.362)	(0.373)	(0.018)	(0.033)
*ER_it_*	−0.003	−0.001	−0.002 **	−0.002 **	0.074	0.127
	(−1.213)	(−0.677)	(−2.046)	(−2.037)	(0.725)	(1.199)
*M_it_*	0.976 ***	0.969 ***	0.968 ***	0.972 ***	0.957 ***	0.907 ***
	(19.385)	(20.529)	(18.777)	(17.775)	(18.307)	(15.283)
*M_it_*ER_it_*		0.024 *		−0.002		−1.294
		(1.824)		(−0.408)		(−1.539)
*W*ER_it_*	0.010 ***	0.003	−0.000	−0.001	−0.056	−0.129
	(3.681)	(1.224)	(−0.427)	(−0.652)	(−0.527)	(−1.135)
*W*M_it_*	0.120	0.055	0.108	0.102	0.094	0.133
	(1.136)	(0.542)	(1.014)	(0.951)	(0.886)	(1.232)
*W*M_it_*ER_it_*		−0.122 ***		−0.008		1.577
		(−5.536)		(−0.905)		(1.498)
Control variables	Yes	Yes	Yes	Yes	Yes	Yes
R-squared	0.675	0.803	0.520	0.522	0.569	0.586
N	187	187	187	187	187	187

Note: (1) z-statistics in the parentheses; (2) ***, **, and * denote *p* < 0.01, *p* < 0.05, and *p* < 0.1, respectively.

**Table 9 ijerph-18-11449-t009:** The moderating results of efficiency change in the eastern region.

Variables	Command–Control		Market-Incentive		Public-Participation	
	(1)	(2)	(3)	(4)	(5)	(6)
*ML_i,t−_* _1_	−0.076	−0.079	−0.083	−0.071	−0.092 *	−0.097 *
	(−1.480)	(−1.531)	(−1.618)	(−1.363)	(−1.739)	(−1.836)
*W*ML_it_*	0.436 ***	0.405 ***	0.440 ***	0.424 ***	0.445 ***	0.445 ***
	(7.849)	(7.053)	(7.973)	(7.607)	(8.082)	(8.066)
*ER_it_*	−0.002	−0.001	0.002	0.002	0.069	0.121
	(−0.629)	(−0.198)	(1.405)	(1.272)	(0.456)	(0.758)
*M_it_*	0.928 ***	1.016 ***	0.888 ***	0.991 ***	0.881 ***	0.848 ***
	(8.393)	(8.669)	(8.242)	(7.792)	(8.337)	(7.616)
*M_it_*ER_it_*		−0.100 **		−0.022		1.779
		(−2.179)		(−1.535)		(0.853)
*W*ER_it_*	−0.004	−0.005	−0.003 *	−0.003 **	−0.231	−0.310
	(−1.001)	(−1.202)	(−1.802)	(−2.107)	(−1.486)	(−1.465)
*W*M_it_*	−0.234	0.052	−0.465 **	−0.119	−0.406 **	−0.374 *
	(−1.057)	(0.202)	(−2.137)	(−0.437)	(−1.982)	(−1.751)
*W*M_it_*ER_it_*		−0.207 **		−0.051 **		−1.475
		(−2.240)		(−2.162)		(−0.465)
Control variables	Yes	Yes	Yes	Yes	Yes	Yes
R-squared	0.202	0.361	0.241	0.279	0.194	0.193
N	187	187	187	187	187	187

Note: (1) z-statistics in the parentheses; (2) ***, **, * denote *p* < 0.01, *p* < 0.05, and *p* < 0.1, respectively.

**Table 10 ijerph-18-11449-t010:** The moderating results of technical change in the central and western regions.

Variables	Command–Control		Market-Incentive		Public-Participation	
	(1)	(2)	(3)	(4)	(5)	(6)
*ML_i,t−_* _1_	0.001	0.008	0.004	0.002	−0.002	−0.010
	(0.037)	(0.257)	(0.136)	(0.053)	(−0.062)	(−0.313)
*W*ML_it_*	0.133 **	0.135 **	0.126 **	0.132 **	0.127 **	0.133 **
	(2.395)	(2.438)	(2.258)	(2.384)	(2.292)	(2.403)
*ER_it_*	0.006	0.005	−0.001	−0.001	0.062	0.070
	(1.306)	(1.204)	(−0.918)	(−0.961)	(0.747)	(0.850)
*M_it_*	0.818 ***	0.810 ***	0.818 ***	0.832 ***	0.818 ***	0.786 ***
	(18.796)	(18.387)	(18.281)	(18.530)	(18.740)	(16.109)
*M_it_*ER_it_*		−0.034		0.006 *		−0.636
		(−1.465)		(1.799)		(−0.813)
*W*ER_it_*	−0.009	−0.009	0.000	0.000	0.150	0.148
	(−1.515)	(−1.493)	(0.231)	(0.310)	(0.939)	(0.932)
*W*M_it_*	0.025	0.029	0.039	0.019	0.036	0.135 *
	(0.354)	(0.403)	(0.556)	(0.262)	(0.518)	(1.655)
*W*M_it_*ER_it_*		0.006		−0.006 *		2.733 **
		(0.134)		(−1.688)		(2.323)
Control variables	Yes	Yes	Yes	Yes	Yes	Yes
R-squared	0.580	0.585	0.582	0.588	0.575	0.590
N	323	323	323	323	323	323

Note: (1) z-statistics in the parentheses; (2) ***, **, and * denote *p* < 0.01, *p* < 0.05, and *p* < 0.1, respectively.

**Table 11 ijerph-18-11449-t011:** The moderating results of efficiency change in the central and western regions.

Variables	Command–Control		Market-Incentive		Public-Participation	
	(1)	(2)	(3)	(4)	(5)	(6)
*ML_i,t−_* _1_	−0.047	−0.045	−0.089 **	−0.086 **	−0.047	−0.053
	(−1.159)	(−1.126)	(−2.209)	(−2.105)	(−1.172)	(−1.307)
*W*ML_it_*	0.418 ***	0.419 ***	0.394 ***	0.394 ***	0.417 ***	0.420 ***
	(10.089)	(10.106)	(9.432)	(9.420)	(10.025)	(10.140)
*ER_it_*	−0.002	−0.002	0.001	0.001	−0.008	0.033
	(−0.276)	(−0.337)	(1.329)	(1.191)	(−0.081)	(0.302)
*M_it_*	0.695 ***	0.685 ***	0.708 ***	0.707 ***	0.698 ***	0.697 ***
	(10.172)	(9.824)	(10.688)	(10.649)	(10.261)	(10.306)
*M_it_*ER_it_*		−0.001		0.001		1.279
		(−0.016)		(0.252)		(1.486)
*W*ER_it_*	−0.000	−0.001	0.005 ***	0.005 ***	−0.190	−0.218
	(−0.013)	(−0.106)	(3.922)	(3.187)	(−0.938)	(−1.032)
*W*M_it_*	−0.126	−0.158	−0.091	−0.094	−0.123	−0.135
	(−1.338)	(−1.517)	(−0.988)	(−1.017)	(−1.309)	(−1.336)
*W*M_it_*ER_it_*		−0.057		0.002		−0.859
		(−0.749)		(0.334)		(−0.498)
Control variables	Yes	Yes	Yes	Yes	Yes	Yes
R-squared	0.131	0.125	0.166	0.167	0.143	0.146
N	323	323	323	323	323	323

Note: (1) z-statistics in the parentheses; (2) *** and ** denote *p* < 0.01 and *p* < 0.05 respectively.

## Data Availability

The data used to support the findings of this study are available from the corresponding author upon request.

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
