# Peer review of "Research on the Impacts of Heterogeneous Environmental Regulations on Green Productivity in China: The Moderating Roles of Technical Change and Efficiency Change"

_ijerph, 2021, doi:10.3390/ijerph182111449_

Round 1
Reviewer 1 Report
This paper studied the relationship between different types of environmental regulations on green productivity and also the moderating effects of technical change and efficiency change on this relationship. The work of this paper is clear and it makes several contributions. However, I still have three concerns which needs to be further solved.
- The introduction section should propose and highlight the research issue.
- The conclusion section should add some concise summaries, especially compared with the existing literature.
- Revise the policy recommendations section based on research conclusions, highlight your implications for practice.
Author Response
Responses to Reviewer # 1
Comments and Suggestions for Authors
This paper studied the relationship between different types of environmental regulations on green productivity and also the moderating effects of technical change and efficiency change on this relationship. The work of this paper is clear and it makes several contributions. However, I still have three concerns which needs to be further solved.
The introduction section should propose and highlight the research issue.
Reply: Many thanks for your constructive suggestion, we have rewritten the introduction section and added plenty of materials to highlight the research issue.
The conclusion section should add some concise summaries, especially compared with the existing literature.
Reply: Many thanks for your constructive suggestion, we have followed your advice and added some concise summaries in the conclusion section, especially compared with the existing literature.
Revise the policy recommendations section based on research conclusions, highlight your implications for practice.
Reply: Many thanks for your constructive suggestion, we have followed your advice and revised the policy recommendations section based on research conclusions, and highlighted our implications for practice.
Above are the detailed corrections we made according to your comments point by point. Thanks again for your meticulous review and valuable suggestions to improve our manuscript. We hope that the revised revision has addressed all the issues. We are looking forward to your positive response. If you have any queries, please don’t hesitate to contact me at the address below.
Best regards,
Yanchao Feng 1, Yong Geng 2, Zhou Liang 3,*, Qiong Shen 1 and Xiqiang Xia 1
1 Business School, Zhengzhou University, Zhengzhou 450001, PR China
2 School of Environmental Science and Engineering, Shanghai Jiao Tong University, Shanghai 200240, PR China
3 School of Economics and Management, Harbin Institute of Technology, Harbin 150001, PR China
* Correspondence: liangzhou0805@163.com (Z. Liang)
Reviewer 2 Report
The subject of the study is interesting and original. The contributions of the study to the literature are clearly stated by the authors. The econometric techniques used in the study are explained in detail. In the study, different estimation results are also given comparatively. The literature has been extensively analyzed. A large number of control variables were used in econometric estimations. I found the article generally successful. Authors only need to correct some simple English spelling errors.
Author Response
Responses to Reviewer # 2
Comments and Suggestions for Authors
The subject of the study is interesting and original. The contributions of the study to the literature are clearly stated by the authors. The econometric techniques used in the study are explained in detail. In the study, different estimation results are also given comparatively. The literature has been extensively analyzed. A large number of control variables were used in econometric estimations. I found the article generally successful. Authors only need to correct some simple English spelling errors.
Reply: Many thanks for your positive comments. Even so, we have rewritten this paper and invited two English native experts to improve the academic level of this study, and we hope that the revised revision has addressed all the issues. We are looking forward to your positive response. If you have any queries, please don’t hesitate to deliver your new comments. Best wishes for you and God bless you.
Best regards,
Yanchao Feng 1, Yong Geng 2, Zhou Liang 3,*, Qiong Shen 1 and Xiqiang Xia 1
1 Business School, Zhengzhou University, Zhengzhou 450001, PR China
2 School of Environmental Science and Engineering, Shanghai Jiao Tong University, Shanghai 200240, PR China
3 School of Economics and Management, Harbin Institute of Technology, Harbin 150001, PR China
* Correspondence: liangzhou0805@163.com (Z. Liang)
Reviewer 3 Report
Thank you for an interesting article. Some suggestions for improvement are as follows:
Introduction- While the authors outline that no similar work is done in China, please outline if similar studies are conducted outside china. Also within China what work is done on or around green productivity etc.
Lit Review - Any limitations of the ML index?
Methods - Line 257: Give details of these Yearbooks. How and by whom is this data collected and collated in the Yearbooks?
Results - Results are articulately presented but not discussed at all. This makes the positioning of the article quite weak.
Conclusion - Any limitations and future directions of the study must be included in the conclusion.
Author Response
Responses to Reviewer # 3
Comments and Suggestions for Authors
Thank you for an interesting article. Some suggestions for improvement are as follows:
Introduction- While the authors outline that no similar work is done in China, please outline if similar studies are conducted outside china. Also within China what work is done on or around green productivity etc.
Reply: Many thanks for your constructive suggestion, we have rewritten the introduction section and the literature review section, and compared our research with relevant studies not only in China but also Outside China.
Literature Review - Any limitations of the ML index?
Reply: Indeed, the ML index has its own limitations. For instance, according to the calculation method of [28], the unfeasible solution of the ML index can be merely solved with constant returns to scale, but not with variable returns to scale. However, this is not the main focus of this study, and we deliver this issue in the section of “research prospects”.
Methods - Line 257: Give details of these Yearbooks. How and by whom is this data collected and collated in the Yearbooks?
Reply: Many thanks for your constructive suggestion, we have followed your advice and given details of these Yearbooks. In addition, the work of data collected was done by the author Zhou Liang.
Results - Results are articulately presented but not discussed at all. This makes the positioning of the article quite weak.
Reply: Many thanks for your constructive suggestion, we have followed your advice and given our discussion in the Results section.
Conclusion - Any limitations and future directions of the study must be included in the conclusion.
Reply: Many thanks for your constructive suggestion, we have followed your advice and added our research prospects from four aspects as follows:
This study should also be interpreted with some limitations, which in turn gives some recommendations for the future research. First, due to data restrictions, the provincial panel data is employed in this study. To confirm our findings, the prefecture-level and county-level panel data can be used for comprehensive and thorough analysis. Second, in our study, environmental regulations are divided into three types based on the difference of regulated executors, and each type of environmental regulations is measured by a single indicator in empirical research. In further research, an expansion of the compound indicator system may be considered to obtain more guiding conclusions. Third, according to the calculation method of [28], the unfeasible solution of the ML index can be merely solved with constant returns to scale, but not with variable returns to scale. Thus, more advanced calculation method of efficiency should be employed in the future. Last but not least, the spatial Durbin model is adopted to do the empirical analysis in this paper, but individual effect is ignored. To expand the research, the Geographically and Temporally Weighted Regression (GTWR) model could be adopted to empirically study the impact of environmental regulations on green productivity in China and other developing countries undergoing similar urbanization and industrialization processes.
Above are the detailed corrections we made according to your comments point by point. Thanks again for your meticulous review and valuable suggestions to improve our manuscript. We hope that the revised revision has addressed all the issues. We are looking forward to your positive response. If you have any queries, please don’t hesitate to contact me at the address below.
Best regards,
Yanchao Feng 1, Yong Geng 2, Zhou Liang 3,*, Qiong Shen 1 and Xiqiang Xia 1
1 Business School, Zhengzhou University, Zhengzhou 450001, PR China
2 School of Environmental Science and Engineering, Shanghai Jiao Tong University, Shanghai 200240, PR China
3 School of Economics and Management, Harbin Institute of Technology, Harbin 150001, PR China
* Correspondence: liangzhou0805@163.com (Z. Liang)
Reviewer 4 Report
This manuscript develops a theme already covered in the scientific literature, however it has potential to convert into an original contribution, but some improvements are needed which I report below.
- The abstract does not perfectly follow the structure suggested by the journal guidelines namely: 1) Background: Place the question addressed in a broad context and highlight the purpose of the study; 2) Methods: Describe briefly the main methods or treatments applied. Include any relevant preregistration numbers, and species and strains of any animals used. 3) Results: Summarize the article's main findings; and 4) Conclusion: Indicate the main conclusions or interpretations. Specifically in the Background, adequate arguments are not provided to justify the reasons for this study. It is unclear to the reader what the moderating roles of technical change and efficiency change are, as well as why ignoring them is a problem.
- In the introduction, beginning at line 87, the authors state that this study makes a threefold contribution, but it is not clear what theoretical support these contributions are based on. The authors need to explain what GAPs of recent literature the study's contributions fill.
- In the literature review, the authors should explain how this manuscript differs from the following previously published paper that was not cited:
Environmental Regulation, Green Innovation, and Industrial Green Development: An Empirical Analysis Based on the Spatial Durbin Model
https://doi.org/10.3390/su10010223
- The empirical part is described with rigor and is easy to understand for the reader.
- In the conclusion, authors should include, in addition to policy implications, the limitations of this study
Author Response
Responses to Reviewer # 4
Comments and Suggestions for Authors
This manuscript develops a theme already covered in the scientific literature, however it has potential to convert into an original contribution, but some improvements are needed which I report below.
The abstract does not perfectly follow the structure suggested by the journal guidelines namely: 1) Background: Place the question addressed in a broad context and highlight the purpose of the study; 2) Methods: Describe briefly the main methods or treatments applied. Include any relevant preregistration numbers, and species and strains of any animals used. 3) Results: Summarize the article's main findings; and 4) Conclusion: Indicate the main conclusions or interpretations. Specifically in the Background, adequate arguments are not provided to justify the reasons for this study. It is unclear to the reader what the moderating roles of technical change and efficiency change are, as well as why ignoring them is a problem.
Reply: Many thanks for your constructive suggestion, we have followed your advice and rewritten the Abstract from the four listed aspects: Background, Methods, Results, and Conclusions.
In the introduction, beginning at line 87, the authors state that this study makes a threefold contribution, but it is not clear what theoretical support these contributions are based on. The authors need to explain what GAPs of recent literature the study's contributions fill.
Reply: Many thanks for your constructive suggestion, we have added the theoretical support (i.e., the Porter Hypothesis) and explain the gaps of recent literature that this study fills.
In the literature review, the authors should explain how this manuscript differs from the following previously published paper that was not cited:
Environmental Regulation, Green Innovation, and Industrial Green Development: An Empirical Analysis Based on the Spatial Durbin Model
https://doi.org/10.3390/su10010223
Reply: Many thanks for your constructive suggestion, we have downloaded this paper and added it as one cited research. Indeed, some contents of this paper are similar with our study, while the research topic of these two papers are different. In addition, our study employs more econometric models and delivers more novel findings, which has expanded the former published paper.
The empirical part is described with rigor and is easy to understand for the reader.
Reply: Many thanks for your constructive suggestion, we have rewritten this section and spare no effect to make this section easy to understand for the reader.
In the conclusion, authors should include, in addition to policy implications, the limitations of this study
Reply: Many thanks for your constructive suggestion, we have added the limitations and research prospects in the last section as follows:
This study should also be interpreted with some limitations, which in turn gives some recommendations for the future research. First, due to data restrictions, the provincial panel data is employed in this study. To confirm our findings, the prefecture-level and county-level panel data can be used for comprehensive and thorough analysis. Second, in our study, environmental regulations are divided into three types based on the difference of regulated executors, and each type of environmental regulations is measured by a single indicator in empirical research. In further research, an expansion of the compound indicator system may be considered to obtain more guiding conclusions. Third, according to the calculation method of [28], the unfeasible solution of the ML index can be merely solved with constant returns to scale, but not with variable returns to scale. Thus, more advanced calculation method of efficiency should be employed in the future. Last but not least, the spatial Durbin model is adopted to do the empirical analysis in this paper, but individual effect is ignored. To expand the research, the Geographically and Temporally Weighted Regression (GTWR) model could be adopted to empirically study the impact of environmental regulations on green productivity in China and other developing countries undergoing similar urbanization and industrialization processes.
Above are the detailed corrections we made according to your comments point by point. Thanks again for your meticulous review and valuable suggestions to improve our manuscript. We hope that the revised revision has addressed all the issues. We are looking forward to your positive response. If you have any queries, please don’t hesitate to contact me at the address below.
Best regards,
Yanchao Feng 1, Yong Geng 2, Zhou Liang 3,*, Qiong Shen 1 and Xiqiang Xia 1
1 Business School, Zhengzhou University, Zhengzhou 450001, PR China
2 School of Environmental Science and Engineering, Shanghai Jiao Tong University, Shanghai 200240, PR China
3 School of Economics and Management, Harbin Institute of Technology, Harbin 150001, PR China
* Correspondence: liangzhou0805@163.com (Z. Liang)
Round 2
Reviewer 4 Report
Dear Authors,
I appreciate your efforts to improve your paper following the suggestions of the reviewers, so I consider this latest version of the manuscript suitable for publication.
Congratulations!